# Variation of Chemical Components in Sapwood, Transition Zone, and Heartwood of *Dalbergia odorifera* and Its Relationship with Heartwood Formation

**Ruoke Ma** [1] **, Heng Liu** [1,2] **, Yunlin Fu** [1,]*, **Yingjian Li** [1] **, Penglian Wei** [1] **and Zhigao Liu** [1]

1   College of Forestry, Guangxi University, Nanning 540004, China; 1909401001@st.gxu.edu.cn (R.M.);
    1709302002@alu.gxu.edu.cn (H.L.); liyingjian@gxu.edu.cn (Y.L.); weipenglian@gxu.edu.cn (P.W.);
    lzgk18@gxu.edu.cn (Z.L.)
2   Experimental Center of Tropical Forestry, Chinese Academy of Forestry, Pingxiang 532600, China
*   Correspondence: fuyunlin@gxu.edu.cn

**Abstract:** Heartwood has a high economic value because of its natural durability, beautiful color, special aroma, and richness in active ingredients used in traditional Chinese medicine. However, the mechanism of heartwood formation remains unclear. *Dalbergia odorifera* was selected as the object of research to analyze this variation in the chemical composition of sapwood, transition zone, and heartwood as well as to elucidate the relationship between this variation and the formation of heartwood. The variation of secondary metabolites was analyzed using gas chromatography-mass spectrometry and ultra-high performance liquid chromatography–mass spectrometry, the variation of lignin was analyzed using Fourier transform infrared spectroscopy and ultraviolet visible spectrophotometry, and the variation law of mineral elements was analyzed using atomic absorption spectrophotometry. The results demonstrated that contents of characteristic secondary metabolites in *Dalbergia odorifera* were mainly distributed in heartwood (84.3–96.8%), increased from the outer to inner layers of the xylem, and sudden changes occurred in the transition zone (the fourth growth ring). The *Dalbergia odorifera* lignin can be identified as typical "syringyl–guaiacyl (S–G)" lignin, and the color darkened from the outside to the inside. The results demonstrated that there were more benzene rings and conjugated C=O structures in the heartwood. Additionally, the variation of minerals in the xylem was related to elemental types; the average concentrations of Mg, Ca, Fe and Sr were higher in the heartwood than in the sapwood, whereas the concentrations of K and Zn were higher in the sapwood than in the heartwood owing to the reabsorption of elements. The concentrations of Na and Cu were similar in the heartwood and sapwood. The composition and structural characteristics of secondary metabolites, lignin, and mineral elements in the three typical xylem regions (sapwood, transition zone and heartwood) of *Dalbergia odorifera* changed. The most abrupt change occurred in the narrow xylem transition zone, which is the key location involved in heartwood formation in *Dalbergia odorifera.*

**Keywords:** *Dalbergia odorifera*; sapwood; transition zone; heartwood; secondary metabolites; lignin; mineral elements

## 1. Introduction

In nature, the xylem of most mature trees consist of sapwood, transition zone, and heartwood. Heartwood is usually defined as the inner layer of the xylem, which is dark in color and contains no active cells [1,2], or a location of secondary metabolite accumulation [3]. In comparison with sapwood, heartwood has a high economic value because of its natural durability, beautiful color, special aroma, and richness in active ingredients in traditional Chinese medicine [4]. The transformation of sapwood into heartwood is an extremely complex physiological and biochemical process. It involves changes in the chemical composition of the xylem with the programmed death of parenchyma cells [5–7]. However, the mechanism of heartwood formation still remains unclear.

Heartwood accumulates a significant number of secondary metabolites (woody known as extractives) [8,9]. Based on the description of Kampe and Magel (2013), three patterns of their production are summarized [4]. Type I or Robinia-type involves the heartwood formation in the transition zone, that is, the conversion of the precursor to the secondary metabolite mainly occurs in the transition zone [10]. The heartwood components of type II or Juglans-type trees begin to form in the sapwood and are found in maximum quantity in the transition zone [11,12]. In the type III, or Santalum-type, of trees, the heartwood components are mainly generated in the heartwood itself, that is, there are still a few living cells in the heartwood which are the main sites of formation of heartwood material [8]. After the parenchyma cells are formed, the secondary metabolites migrate to the surrounding vessels, wood fibers, and other tissues [13,14]. The location of secondary metabolites during xylem formation can help determine the key cue to understanding heartwood formation.

The chemical components of the cell wall affect the properties of the wood and thus direct its subsequent utilization [15,16]. The lignin content of the cell wall changes during the transformation of sapwood to heartwood. In the 1970s, Hergert found that the content of Klason lignin in heartwood was always higher than that in sapwood, and first proposed the concept of "secondary lignification" [17]. However, in a subsequent study, it was found that the lignin content of the heartwood, transition zone, and sapwood of *Robinia pseudoacacia* was almost the same. Therefore, Megal et al. believed that the phenomenon of "secondary lignification" during heartwood formation may be due to the accumulation of phenolic substances resulting in the "pseudo lignification" of the cell wall [18]. A subsequent study revealed that the lignin content of the heartwood of *Eucalyptus urophylla* and *Tectona grandis* were lower than that of their corresponding sapwood [16,19]. Generally, research on lignin deposition in the cell wall during heartwood formation is lacking, and it is still uncertain whether the lignin content changes during sapwood to heartwood transformation.

During the transformation from sapwood to heartwood, the translocation of mineral elements lead to differences in the type and content of metal elements in different regions of the xylem [4,7]. Previous studies on Eucalyptus proposed that the concentration of mineral elements in sapwood was always higher than that in heartwood [20]. However, it was found that the concentration of some mineral elements in the sapwood was lower than that in heartwood. Based on the study of *Magnolia officinalis*, *Liriodendron tulipifera*, and *Melia Sinica*, it was found that there are three distribution patterns of mineral elements in trees [21]. The first distribution pattern was a gradual increase in the concentration of mineral elements from the sapwood to heartwood, the second distribution pattern was a gradual decrease in the concentration of mineral elements from the sapwood to heartwood, and the third distribution pattern was that the highest element concentration was present in the transition zone. The type and content of different mineral elements in different xylem regions and the correlation of this data with heartwood formation need to be elucidated.

*Dalbergia odorifera* T. Chen, commonly known as "Huanghuali", is native to Hainan, China. Its heartwood value is very high and it is an ideal material for making high-grade furniture and handicrafts. It is also a valuable traditional Chinese medicine with antibacterial, antiviral, and antitumor effects [22–24]. Since 2007, *D. odorifera* has been planted in large numbers in Fujian, Yunnan, Guangdong, and Guangxi. At present, the artificial cultivation area has exceeded 3500 ha$^2$ [25]. However, the speed of heartwood formation in *D. odorifera* trees is very slow, and no heartwood can be used as yet. This slow growth is the bottleneck facing the artificial cultivation of *D. odorifera*.

In this study, research was conducted using *D. odorifera* as the tree species to explore (1) the position of secondary metabolites in the xylem, (2) the changes in lignin content and structure, and (3) the radial variation of different mineral elements. The relationship between the changes in the chemical composition of sapwood, transition zone, and heartwood as well as heartwood formation was analyzed.

## 2. Materials and Methods

### 2.1. Plant Material

A 10-year-old *D. odorifera* tree, growing in the campus of Guangxi University in Nanning city (22°50′49″ N, 108°17′29″ E), China, was used in this study. Sample was collected in early October 2018. After cutting down the trees with a chain saw, 5 cm-thick basal wood disks were divided into strip and frozen in dry ice immediately. The wood materials were transferred back to laboratory and stored in a freezer at −80 °C until use.

The wood strip was divided into 10 parts with the growth ring as the boundary in Figure 1. Because the fourth growth ring contains both heartwood and transition zone, the wood should be carefully separated based on the boundaries defined by color. Eleven samples were numbered from inside to outside, namely the heartwood (HW; 1, 2, 3), transition zone (TZ; 4X, 4B), and sapwood (SW; 5, 6, 7, 8, 9, 10). The wood was cut into small sticks, freeze-dried, and ground into wood powder through liquid nitrogen grinding before use.

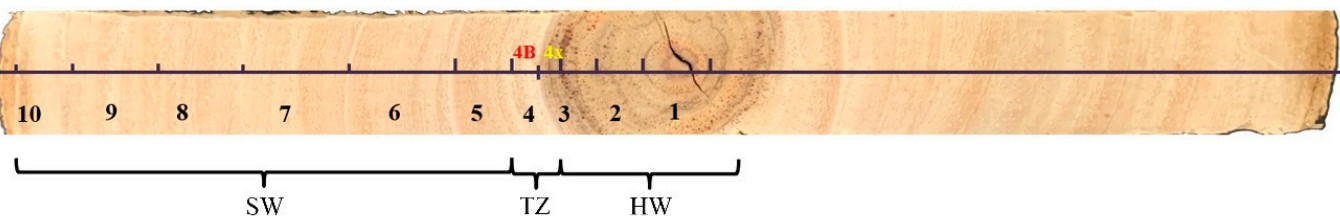

**Figure 1.** Localization of different parts of wood strip with the growth ring as the boundary.

### 2.2. Secondary Metabolites Analysis

#### 2.2.1. Sample Extraction

First, 0.20 g of dried wood powder was accurately weighed into a conical flask, and 10 mL methanol was added to it. Thereafter, ultrasonic extraction was performed for 2 h and this process was repeated three times. Subsequently, the extracts were filtered, combined, and evaporated to dryness through rotary evaporation under vacuum. Then, 10 mL methanol was added for re-dissolution, and it was filtered through a 0.22 μm membrane filter. The extract was diluted appropriately as per detection requirements.

Appropriate amounts (10 mg) of trans-nerolidol, naringenin, isoliquiritigenin, dalbergenin, and formononetin (Shanghai Yuanye Biotechnology Co., Ltd., Shanghai, China) were dissolved in methanol, and the volume was fixed at 10 mL. The purities of all standards were above 98%. Standard mixtures with known concentrations were prepared using the external standard method; the standard solution was diluted to the required concentration of the standard curve, and this solution was measured thrice to establish the standard curve. Seven mixed standard solutions with different concentrations were prepared using the external standard method, and the standard curve was established according to the ratio of peak area to standard concentration. The detection limit was taken as the concentration of the sample when the signal-to-noise ratio (S/N) was 3, while the limit of quantification was taken as the concentration of the sample when the signal-to-noise ratio (S/N) was 10.

#### 2.2.2. Identification of Flavonoids by Ultra-High Performance Liquid Chromatography-Mass Spectrometry/Mass Spectrometry (UPLC-MS/MS)

Flavonoids (naringenin, isoliquiritigenin, dalbergenin, and formononetin) were analyzed by ultra-high performance liquid chromatography–tandem mass spectrometry (UPLC-MS/MS, Waters Corp., Milford, MA, USA) equipped with ACQITY™ UPLC BEH C18 column (2.1 mm × 50 mm, 1.7 μm film thickness, Waters Corp., Milford, MA, USA). The mobile phase was 0.1% formic acid water (A) and 50% methanol acetonitrile solution (B). The elution gradient was as follows: 0–5 min, 10–20% B, 5–10 min, 20–50% B, 10–15 min, 50–90% B, 15–16 min, 90–10% (B) The flow rate was set at 0.4 mL/min, the column temperature at 40 °C and the injection volume at 0.2 μL. Mass spectrometry was performed

with an electrospray ionization ion (ESI) source in positive ion mode and MRM(multiple reaction monitoring) scanning mode. The mass range of measured ions was 50–1200 *m/z* and the corrected ion mass ratio was 556.2771.

### 2.2.3. Gas Chromatography–Mass Spectrometry (GC-MS)

The trans-nerolidol was analyzed with an Agilent 5975C gas chromatography–mass spectrometer (GC-MS, Agilent Technologies Inc., Palo Alto, CA, USA) equipped with HP-5 column (30 m × 0.25 mm, 0.25 μm film thickness). Helium was the carrier gas at a flow rate of 1.0 mL/min with an ionization voltage of 70 eV. The ion source temperature was 250 °C. Samples were diluted by methanol and then injected (1 μL) automatically using split ratio (10:1). The column temperature was initially raised from 60 °C to 160 °C at a rate of 10 °C $\min^{-1}$, and then gradually increased to 220 °C at a rate of 4 °C $\min^{-1}$. Mass range increased from 50 to 500 *m/z* and the solvent delay time was 3 min.

### 2.2.4. Methodology Validation

Precision test: Intra-day precision was obtained by measuring six samples after the same treatment within 24 h, whereas inter-day precision was obtained by measuring the same sample for 72 consecutive hours. The recovery test was performed by adding a known amount of mixed standard solution (80%, 100% and 120%) to a sample with known content. Stability tests were performed by storing the sample extract in a refrigerator at 4 °C and conducting measurements every 12 h for 48 h.

### 2.3. *Preparation and Analysis of Lignin*

Characterization and analysis of lignin in the milled lignin obtained from wood were performed using the Beckman method [26]. First, dried wood flour (4.0000 g) was accurately weighed and extracted with benzene/ethanol mixed solution (volume ratio 2:1) in a Soxhlet extractor until the extract solution was colorless. Thereafter, the solution was dried at low temperature (50 °C) and subsequently milled by planetary ball mill (Hunan Fukasi Experimental Instrument Co., Ltd., Hunan, China) for 4 h.

Next, 40 mL of dioxane aqueous solution (water: dioxane = 4:96) was added. After magnetic stirring in the dark for 24 h, the supernatant was collected through centrifugation. We repeated the above operation three times. After filtration, the extracts were combined. After the dioxane was concentrated under vacuum, the crude lignin was obtained after freeze drying. Then, 5 mL acetic acid solution (90%) was added to the crude lignin and the solution was slowly added dropwise into water (100 mL) with magnetic stirring. After centrifugation, the supernatant was poured out, and the lignin was cleaned with ultrapure water several times until the acetic acid was removed after which it was freeze-dried. Subsequently, 18 mL of dichloroethane/ethanol mixture (volume ratio 2:1) was added, and after the lignin dissolved completely the mixture was centrifuged to remove any insoluble solids. The supernatant was slowly poured into 200 mL anhydrous ether to precipitate lignin. The precipitated lignin was then washed thrice with anhydrous ether and freeze-dried to afford purified ground lignin.

Ultraviolet–visible (UV–Vis) spectrum analysis: 5.0 mg of the lignin sample was accurately weighed into a 10 mL volumetric flask, and a 95% dioxane aqueous solution was added to prepare 0.5 mg/mL lignin solution. Before the test, 1 mL of lignin solution was diluted to 10 mL with a 50% dioxane aqueous solution, and the resulting solution was analyzed using a UV spectrophotometer (Shanghai Spectrum Instruments CO., Ltd., Shanghai, China). The analysis was performed in the measurement range of 0–600 $cm^{-1}$. The experiment was repeated three times.

Infrared spectrum (Thermo Fisher Scientific, Waltham, MA, USA) analysis: The lignin sample was carefully ground, 1–2 mg of the sample was ground with 0.1 g potassium bromide, and the mixture was pressed into a tablet. The analysis was performed in the measurement range of 500–4000 $cm^{-1}$ with 32 cumulative scans; the OPD speed was

0.2 cm/s, the resolution was 4 cm$^{-1}$, and the KBr interference was deducted in real time. The experiment was repeated three times.

### 2.4. Determination of Mineral Elements in D. odorifera Wood

2.4.1. Sample Pretreatment by High-Temperature Ashing

Determination of mineral elements in *D. odorifera* wood was performed based on a previous study [27]. Wood flour (2.00 g) was accurately weighed into a crucible, and placed in a muffle furnace at 600 °C for 6 h. Subsequently, 5 mL of hydrochloric acid solution (50%) was added and the solution was heated until it nearly evaporated completely. After dissolving the remaining solution with 5% hydrochloric acid solution, the resultant solution was transferred into a 10 mL volumetric flask, and the volume was determined with ultrapure water.

2.4.2. Establishment of Standard Curves

Standard stock solutions of K, Cu, Zn, Fe, Na, Ca, Mg and Sr were used to prepare five gradient working solutions of known concentrations (0–2.5 mg/mL). The Na standard working solution contained 0.1 mL potassium oxide (3.8%) solution, whereas the dilution of Ca, Mg and Sr standard working solutions contained 0.1 mL lanthanum oxide solution. The samples were analyzed using an atomic absorption spectrophotometer (aa-7000, Shimadzu, Kyoto, Japan) (the instrument parameters are listed in Table 1) and the standard working curve of each element was established.

**Table 1.** Instrument parameter of atomic absorption spectrophotometer.

|  | K | Na | Mg | Ca | Fe | Cu | Zn | Sr |
|---|---|---|---|---|---|---|---|---|
| Detection wavelength/nm | 766.5 | 589.0 | 285.2 | 422.7 | 248.3 | 324.8 | 213.9 | 460.7 |
| channel width/nm | 0.5 | 0.2 | 0.5 | 0.5 | 0.2 | 0.5 | 0.5 | 0.5 |
| lamp current/mA | 10 | 12 | 8 | 10 | 12 | 6 | 8 | 8 |
| flame type | air-acetylene | | | | | | | |

## 3. Results and Discussion

### 3.1. Radial Variation of Flavonoids

To analyze the radial variation of the secondary metabolites in the xylem of *D. odorifera*, quantitative methods for characteristic flavonoids and terpenoids were established and validated. The quantitative method established for the determination of naringenin, isoliquiritigenin, dalbergenin, and spinononetin was based on UPLC-MS, meanwhile the method for trans-nerolidol used GC-MS. The five compounds investigated showed a good linear variation in the standard curve range ($R^2 > 0.99$), as well as a low limit of detection (LOD) and limit of quantification (LOQ), indicating the high sensitivity of the instrument (Table 2). Because the instruments used to detect terpenoids and flavonoids are different, the LOD and LOQ of trans-nerolidol are higher than those of the flavonoids; however, this does not affect detection as the concentration of the detected samples is above the limit of quantification. The intra- and inter-day precision results of the compounds are shown in Supplemental Table S1. The intra- and inter-day relative standard deviations (RSDs) of the compounds were less than 5%, indicating the reproducibility of the method. Samples with known concentrations were added to a known amount of the mixed standard solution, and subsequently extracted and analyzed using UPLC-MS-MS or GC-MS. As shown in Supplemental Table S2, the recoveries of the compounds were all between $100 \pm 10\%$, and the RSD values were less than 5%, indicating the accuracy of the method. The mixed standard and sample extraction solutions were stored in a refrigerator at 4 °C for 48 h. The results demonstrated that the RSD values of the mixed flavonoid standard and sample extraction solutions were less than 5%, indicating that these solutions were stable after 48 h at 4 °C (Supplemental Table S3). Based on these results, the reliability of the quantitative

methods used to analyze naringenin, isoliquiritigenin, dalbergenin, formononetin and trans-nerolidol in *D. odorifera* was confirmed.

**Table 2.** Regression equations for characteristic secondary metabolites.

| Compounds | Standard Curve | Regresion Coefficient ($r^2$) | Rang (µg/L) | LOD (µg/L) | LOQ (µg/L) |
|---|---|---|---|---|---|
| naringenin | y = 57.634x − 502.57 | 0.999 | 10.5–1050 | 1.05 | 3.47 |
| isoliquiritigenin | y = 110.69x − 570.95 | 0.999 | 10–1000 | 0.48 | 1.44 |
| dalbergin | y = 303.38x + 2729.4 | 0.998 | 9–900 | 0.9 | 2.97 |
| formononetin | y = 334.16x + 1198.6 | 0.998 | 8.5–850 | 0.42 | 1.39 |
| trans-nerolidol | Y = 996.1x − 379.54 | 0.999 | 2.97–59.37 | 59.4 | 196.02 |

Flavonoids and terpenoids are the main secondary metabolites in the heartwood of *D. odorifera* [22]. Flavonoids are generally considered the as main source of color in mahogany heartwood [28]. In *D. odorifera*, the percentage composition of the four characteristic flavonoid compounds exhibited an overall increasing trend from sapwood (the 6–10th growth ring) to heartwood (the 1st–3rd growth ring), and abrupt changes occurred in the transition zone (the 4th growth ring, Figure 2). The amount of these flavonoids in the heartwood accounted for 84.3–94.3% of the total content in the xylem, while that in the transition zone accounted for 5.0–11.8% and that in the sapwood accounted for 0.03–4.4%. Terpenoids are mainly responsible for the aroma of mahogany, and one of the characteristic volatile components of *D. odorifera* is trans-nerolidol. The distribution pattern of trans-nerolidol in the xylem was the same as that of flavonoids. The heartwood had the highest content of trans-nerolidol, accounting for approximately 96.8% of the total, while the transition zone and sapwood accounted for approximately 2.6% and 0.5%, respectively.

After the secondary metabolites are produced in the xylem parenchyma cell, they are transported to the surrounding wood fibers and vessels through the ectoplast system [13], and the distribution of the secondary metabolites in the xylem is worth discussing. The radial distribution of secondary metabolites in the xylem can be explained by two approaches. The first approach is that the secondary metabolites are first synthesized and then transported to the inner layer of the xylem; the sapwood of *D. odorifera* already generates a small quantity of secondary metabolites, which are then transported to the inner layer of the xylem through wood rays. It may belong to the type II (*Robinia* type). Or the parenchymal cells in the transition zone synthesize a large number of secondary metabolites after receiving a certain signal, and then transport them to the surrounding tissues, which is similar to type I (*Juglans* type). The second approach is that the secondary metabolites are synthesized in situ, implying that there may be a small number of living cells in the heartwood. The parenchyma cells in the transition zone are the initial sites stimulated by some signal molecules and subsequently synthesized in large numbers in the heartwood. After the partial synthesis of the secondary metabolites in the transition zone, a significant amount of these secondary metabolites is then distributed in the heartwood, and the content still increases with different growth rings. The distribution patterns of secondary metabolites in *D. odorifera* and sandalwood were similar [8]. According to the distribution of secondary metabolites, the secondary metabolite formation pattern of *D. odorifera* is more likely to belong to type III, also known as the "Sandalwood type." However, whether the secondary metabolites in the heartwood are synthesized in situ or transported from the production site, requires further exploration. The subsequent experiments further explore the radial variation in the activities of enzymes related to secondary metabolism in the parenchyma cells in the xylem.

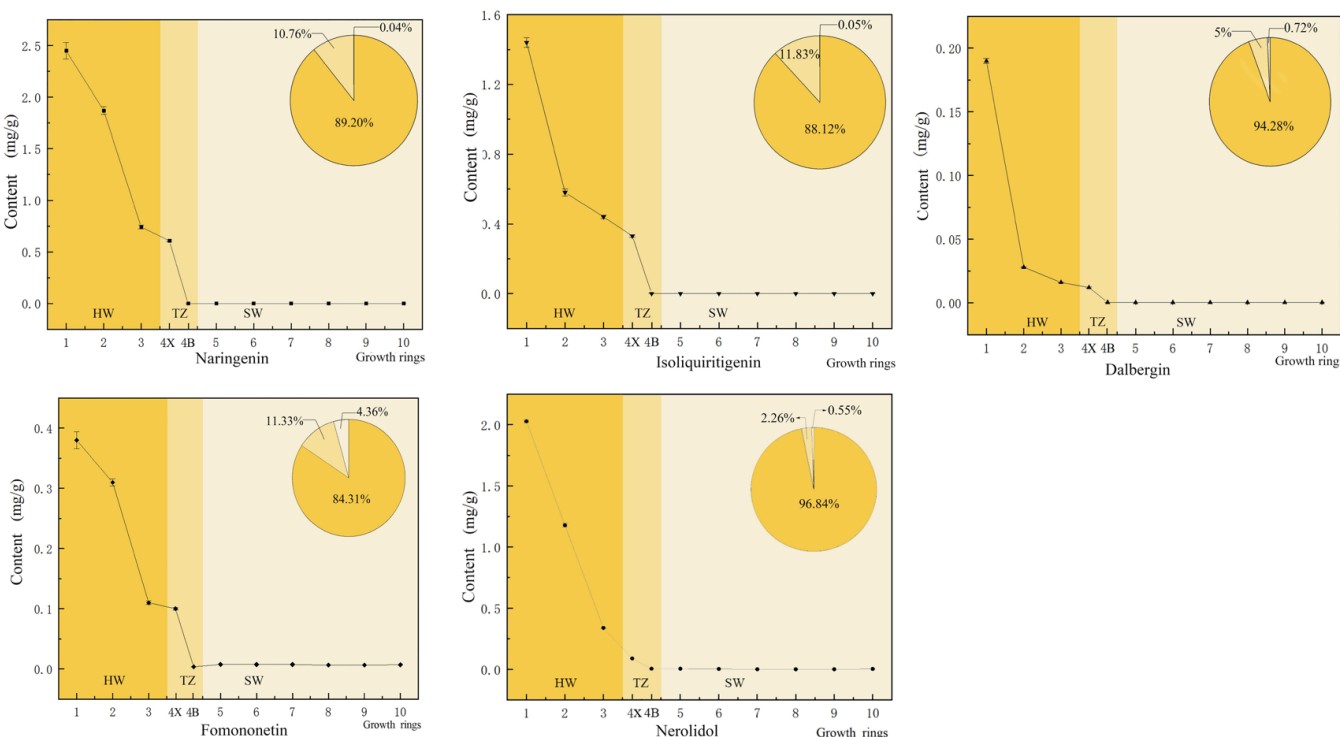

**Figure 2.** Changes in the radial distribution of naringenin, isoliquiritigenin, dalbergenin, formononetin and trans-nerolidol. The horizontal axis shows the change of the growth ring from the pith (the first growth ring) to the outermost sapwood (the tenth growth ring). ▮ heartwood (HW; 1, 2, 3), ▮ transition zone (TZ; 4X, 4B), and ▮ sapwood (SW; 5, 6, 7, 8, 9, 10).

### 3.2. Radial Variation of Lignin

During the transformation of sapwood into heartwood, the changes in the lignin in parenchyma cells directly affect how the wood can be utilized. In addition, the color of lignin differs in different parts of the xylem of *D. odorifera* (Supplemental Figure S1). The color of lignin in the outer layer of sapwood and the transition zone was a lighter cream color, while the color of lignin from the inner layer of the transition zone to the outer layer of the heartwood (the third ring) was dark brown. The color of lignin in the inner layer of the heartwood was dark brown and almost black. In addition, based on the results listed in Supplemental Table S4, the radial variation of the lignin extraction rate in *D. odorifera* is not clear, but in general, the average value of the lignin extraction rate of heartwood appears to be higher than that of sapwood. On the one hand, the deposition process of lignin in sapwood is continuous. On the other hand, there are trace residues or extracts in the early extraction process. After being conjugated with lignin, they were extracted together with lignin, which was also a possible reason for the darker color of lignin in the heartwood.

The UV–Vis spectra of lignin from different growth rings are shown in Figure 3. The maximum absorption peak of all lignin appeared around 236 nm, which was the characteristic absorption of lignin benzene ring [29]. There was a medium intensity absorption band at 274–280 nm, which is the absorption peak of lignin syringyl group [29], indicating that *D. odorifera* lignin is mainly syringyl group, which is consistent with the conclusion of the infrared spectrum (Figure 4). The characteristic absorption of syringae increased gradually from the outer layer of sapwood (the 9th growth ring) to the inner layer of heartwood, which was supposed to be the result of the continuous lignification process of xylem parenchyma cells. In addition, the UV–Vis spectrum of lignin in the outermost layer (tenth growth ring) of the sapwood was significantly lower than that of other growth rings, which may be because the wood was too young and in the initial stage of lignin deposition. The change in the absorption maxima in the UV–Vis spectrum of lignin is consistent with the changes in lignin color, which is due to the unsaturated structure of the benzene ring.

Benzene easily forms a conjugated system responsible for the color of wood and can also be connected with an auxiliary color group to further deepen the wood color [30].

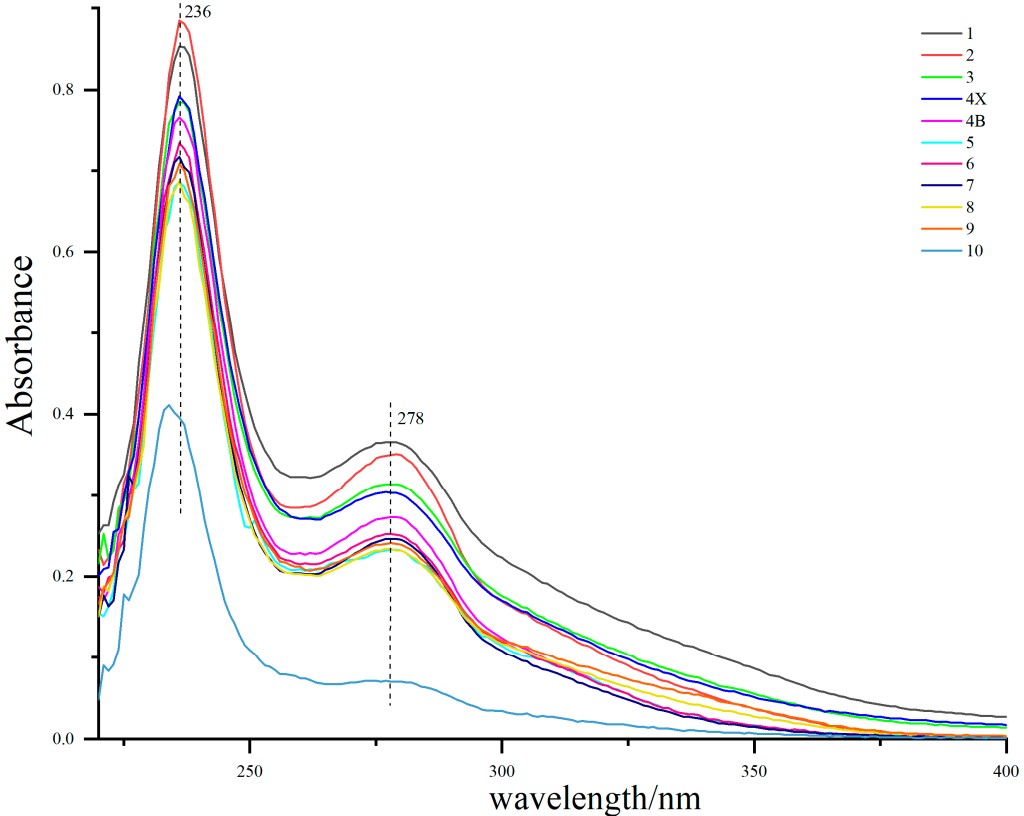

**Figure 3.** Ultraviolet (UV)-visible spectra lignin samples from different growth rings. The growth ring from the pith (the first growth ring) to the outermost sapwood (the tenth growth ring) were distinguished by different colors. Heartwood (HW; 1, 2, 3), transition zone (TZ; 4X, 4B), and sapwood (SW; 5, 6, 7, 8, 9, 10).

The infrared spectra of lignin from different growth rings shown in Figure 4 are similar and show absorption bands at 1603, 1506, 1467, and 1423 cm$^{-1}$ corresponding to the vibration peaks of the aromatic ring carbon skeleton of lignin [31]. The most intense absorption peaks for all types of lignin appeared at 1129 cm$^{-1}$, which were ascribed to the C–O vibration of Syringa group like the peaks at 1216 cm$^{-1}$ and 1328 cm$^{-1}$ [32]. The absorption peaks at 1272 cm$^{-1}$ and 1030 cm$^{-1}$ corresponded to guaiacyl vibrations; this indicated that *D. odorifera* lignin was mainly composed of Syringa-based units and a small number of guaiacyl units and thus, the type of lignin was syringyl–guaiacyl (S–G) lignin. In addition, the stretching vibration absorption peaks of methyl and methylene were present at 3449 cm$^{-1}$, 2944 cm$^{-1}$, and 2843 cm$^{-1}$, respectively, while the non-conjugated C=O stretching vibration absorption peak was present at 1728 cm$^{-1}$ [33]. The most significant change in the infrared spectra of lignin from different growth rings was observed at 1671 cm$^{-1}$, which was attributed to the stretching vibration of conjugated C=O; this peak was absent in the sapwood samples. However, the presence of the conjugated C=O based structure indicates more chromogenic groups in lignin [34] which is one of the reasons why lignin from the heartwood of *D. odorifera* had a darker color than the sapwood and transition zone.

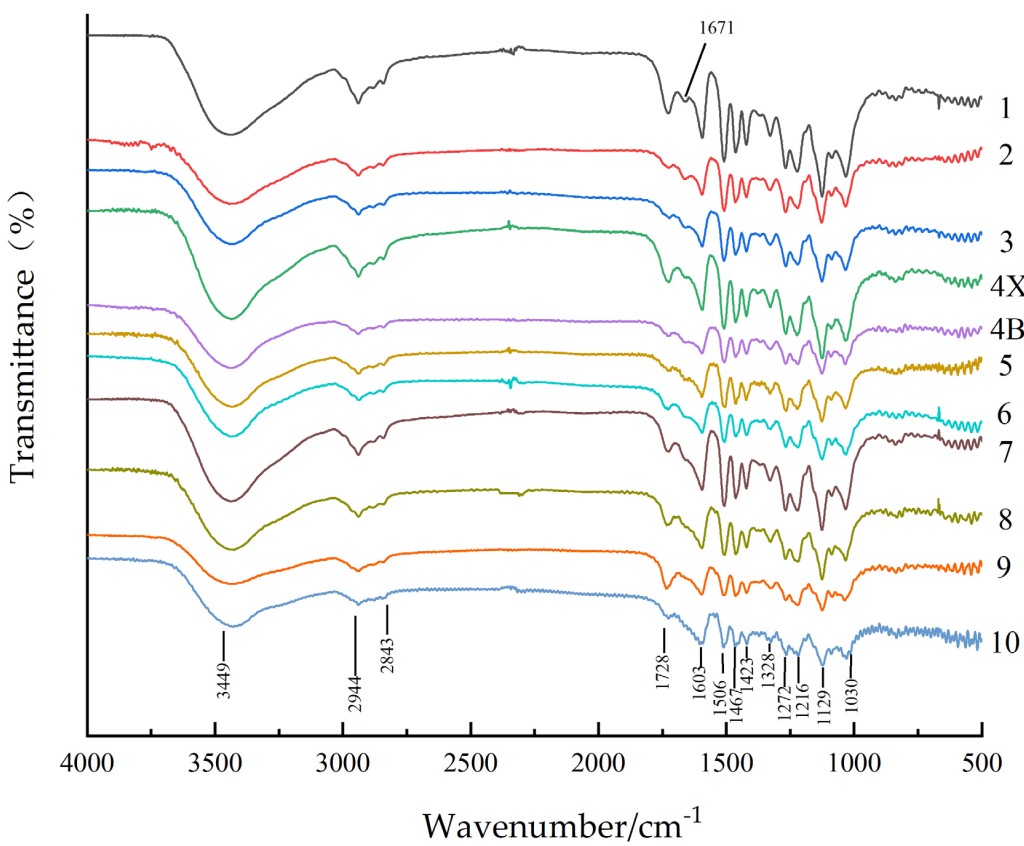

**Figure 4.** Infrared spectra of lignin samples from different growth ring. The growth ring from the pith (the first growth ring) to the outermost sapwood (the tenth growth ring) were distinguished by different colors. Heartwood (HW; 1, 2, 3), transition zone (TZ; 4X, 4B), and sapwood (SW; 5, 6, 7, 8, 9, 10).

*3.3. Radial Variation of Element Distribution in D. odorifera Wood*

The results of analyzing the mineral nutrient elements in different growth rings of *D. odorifera* wood are listed in Figure 5 and show that Ca, K and Mg are the elements with the highest concentrations, with Ca having the highest concentration of 10,193.63 mg/kg, followed by Na and Fe. The concentrations of Zn, Cu and Sr were relatively low, and the trend in variation of mineral elements obtained by atomic absorption spectrophotometry was analyzed (Supplemental Table S5). According to Figure 5, the eight elements can be divided into three categories, among which Mg, Ca, Fe, Sr and other elements can be divided into one group. The average concentrations of Mg, Ca, Fe and Sr in heartwood are higher than those in sapwood. The reason for this phenomenon may be because when the concentration of elements exceeds a certain limit, plants transport the excess elements from the sapwood to the heartwood to maintain optimal levels of growth and development. The elements K and Zn belong to the second category, and their concentration in the sapwood is higher than that in the heartwood. K participates in the activation of enzymes, while Zn plays an important role in sapwood metabolism [35,36]. It is speculated that K and Zn can be reabsorbed by sapwood cells through radial transport. The concentrations of Na and Cu, which belong to the third-category, are similar in the heartwood and sapwood, but the concentration of Na in the transition zone was higher. There was no clear variation in Cu content in the entire xylem or the radial direction. The distribution of K, Ca, and Sr in the xylem of *D. odorifera* was similar to that of K in *Afzelia xylocarpa* [21]. The results demonstrate that the distribution of Na, Mg, Ca, Fe, Zn and Sr in the heartwood and transition zone exhibited clear changes. For instance, the concentration of K decreases sharply in the heartwood, whereas the concentrations of Mg and Ca increase in the heartwood. This

indicates that the change in element concentration at the junction of the heartwood and transition zone is closely related to the mechanism of heartwood formation.

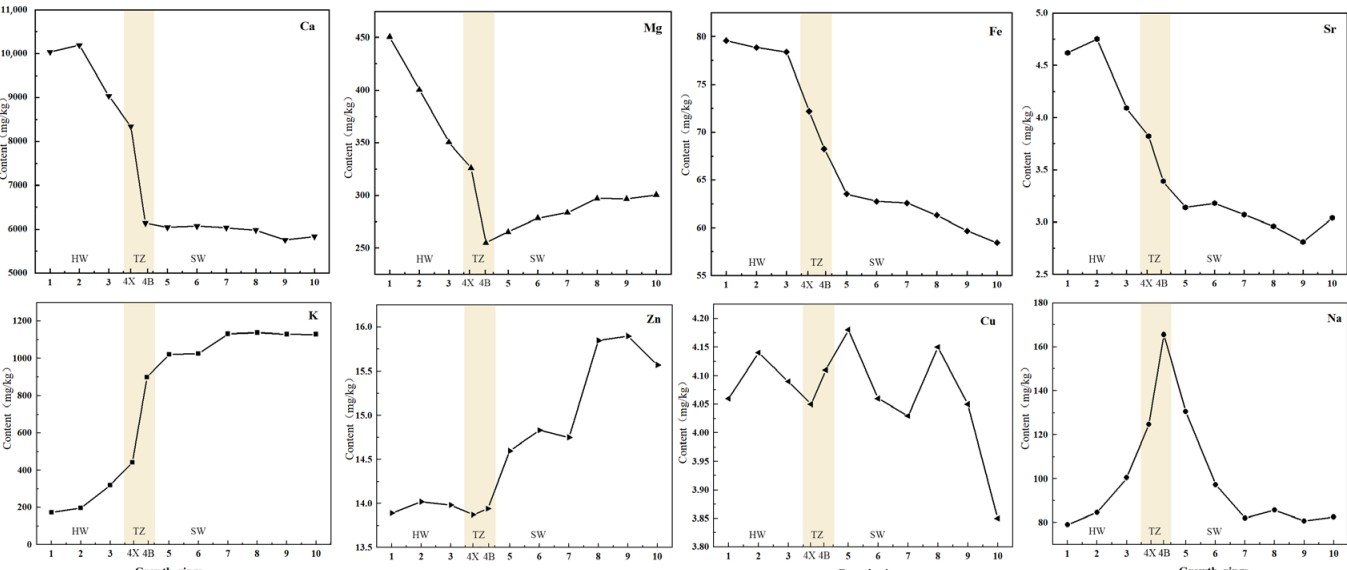

**Figure 5.** Radial variation of mineral elements (Ca, Mg, Fe, Sr, K, Zn, Cu, Na) along the growth ring. Mineral elements are marked in the upper right corner. The horizontal axis shows the change of the growth ring from the pith (the first growth ring) to the outermost sapwood (the tenth growth ring). heartwood (HW; 1, 2, 3), transition zone (TZ; 4X, 4B), and sapwood (SW; 5, 6, 7, 8, 9, 10).

## 4. Conclusions

In this paper, the composition and structural changes of secondary metabolites, lignin, and mineral nutrients in the sapwood, transition zone, and heartwood of *D. odorifera* wood were compared to determine the relationship between the radial variation in chemical composition and the formation of heartwood. The results demonstrated that the flavonoids and terpenoids in *D. odorifera* were mainly distributed in heartwood (84.3–96.8%), and the content increased suddenly in the transition zone. The color of lignin gradually deepened from the outer sapwood to the pith core because of its abundance in benzene rings and conjugated C=O groups. Among the eight mineral elements detected in *D. odorifera*, Ca, K and Mg are major elements, and Ca having the highest concentration of 10,193.63 mg/kg. The change in element concentration at the junction of heartwood and transition zone was found to be closely related to heartwood formation. In conclusion, the production of secondary metabolites (flavonoids and terpenoids), lignin structure, and mineral nutrient content of *D. odorifera* change abruptly in the transition zone. The changes in several chemical composition indices indicate that certain physiological phenomena occur in the transition zone that are directly related to heartwood formation.

**Supplementary Materials:** The following are available online at https://www.mdpi.com/article/1 0.3390/f12050577/s1, Figure S1: The color change of extracted lignin, Table S1: Intra- and inter-day repeatability of five compounds, Table S2: Recoveries of five compounds, Table S3: Stability of five compounds, Table S4: Change of lignin extraction rate, Table S5: Element content of different growth ring in *Dalbergia odorifera*.

**Author Contributions:** Y.F., H.L. and R.M. conceived and designed the experiments, R.M. wrote the paper, H.L. and R.M. performed the experiments and analyzed the data. Y.L., P.W. and Z.L. participated in and help to complete the experiments. All authors have read and agreed to the published version of the manuscript.

**Funding:** This research was supported by the National Natural Science Foundation of China (31870540).

**Institutional Review Board Statement:** Not applicable.

**Informed Consent Statement:** Not applicable.

**Data Availability Statement:** The data presented in this study are available in Supplementary Material Table S5.

**Conflicts of Interest:** The authors declare no conflict of interest.

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
