# Peer review of "Variation of Chemical Components in Sapwood, Transition Zone, and Heartwood of Dalbergia odorifera and Its Relationship with Heartwood Formation"

_forests, doi:10.3390/f12050577_

Round 1
Reviewer 1 Report
The present paper is aimed for evaluation of some chemical compounds in sapwood, transition zone, and heartwood of Dalbergia odorifera.
The paper is interesting, however, I have same suggestions for the improvement of manuscript:
- add some exact results both in abstract and conclusions.
- please, clarify sampling process. I am not familiar with the distribution of phenolic compounds in the stem of tree, but I believe, it should be a difference between the part near ground, and that top one. From which position you took the ring for analysis? If I understand well, you analyzed only one ring... Are you sure, that this ring represents full stem? For precision test, you took samples from one ring, or from different rings?
- other comments is marked in the attached file.

Author Response
1、add some exact results both in abstract and conclusions.
Re: Thank you for the suggestion. We have added some exact results in abstract and conclusions (in P1, line 18; P10, line 382, 386).
2、please, clarify sampling process. I am not familiar with the distribution of phenolic compounds in the stem of tree, but I believe, it should be a difference between the part near ground, and that top one. From which position you took the ring for analysis? If I understand well, you analyzed only one ring... Are you sure, that this ring represents full stem? For precision test, you took samples from one ring, or from different rings?
Re: Thank you for your suggestion.
First of all, let me briefly describe the sampling process:
(1) After cutting down the trees, take the basal disk (P3, line 7 and 8 Modified), and make the disk into strips (Fig.1) for sampling;
(2)The xylem was subdivided into 10 small pieces from the outside to the inside according to the annual rings;
(3) Freeze dry the small pieces and grind them into powder for chemical composition analysis.
Inclusion, we focused on the radial changes of xylem from the outer sapwood to the pith, including different growth rings.
The secondary growth of woody plants experienced the proliferation and differentiation of vascular cambium, the expansion of cells, the formation and lignification of secondary cell walls, and the programmed death of cells, and the formation of heartwood extraction (or namely secondary metabolite) in the last stage of development [1,2]. According to the different life cycle time, the xylem cells of trees can be divided into two types. One is short-lived cell, including fiber and vessel element (tracheid and vessel). Among them, the vessel element died within two days after the differentiation of cambium, and the survival cycle of wood fiber and tracheid cells was slightly longer, usually died within 1 month [3, 4]. The other is long-term cell, mainly parenchyma cells with a life cycle of several or even decades, including axial and ray parenchyma cells. The ultrastructural changes during programmed death of parenchyma in the xylem play an important role in heartwood formation [5-8].
According to the previous literature, we can conclude that the formation of heartwood substance such as phenols is accompanied with the aging of ray or axial parenchyma cells. Because the process of tree thickening is also the process of inner sapwood cell aging. It is important to note that the radial changes of xylem provide a dynamic place for the study of the production of secondary metabolites in parenchyma cells. Therefore, this is the main reason why we choose the radial change of chemical composition in this study.
As for the law of the distribution of phenols in the axial direction, we think it is a good scientific problem. Because trees are a whole with life, radial changes can only explain some laws. We think your novel suggestions can be a good point for our future research.
Reference
[1] Dejardin, A.; Laurans, F.; Arnaud, D.; Breton, C.; Leple, J.C. Wood formation in Angiosperms. Comptes Rendus Biologies 2010, 333, 325-334.
[2] Plomion, C.; Leprovost, G.; Stokes, A. Wood formation in trees. Plant Physiology, 2001, 127, 1513-1523.
[3] Bollhöner, B., Prestele,J., Tuominen, H. Xylem cell death: emerging understanding of regulation and function. Journal of experimental botany 2012, 63, 1081-1094.
[4] Courtois-Moreau, C.L.; Pesquet, E.; Sjödin, A.; MuñIz, L.; Bollhöner, B.; Kaneda, M.; et al. A unique program for cell death in xylem fibers of populus stem. Plant Journal 2010, 58, 260-274.
[5] Frey-Wyssling, A.; Bosshard, H.H. Cytology of the ray cells in sapwood and heartwood. Holzforschung-International journal of the biology, Chemistry, Physics and Technology of Wood 1959, 13, 129-137.
[6] Nakaba, S.; Begum, S.; Yamagishi, Y.; Jin, H. O.; Kubo, T.; & Funada, R. Differences in the timing of cell death, differentiation and function among three different types of ray parenchyma cells in the hardwood populus sieboldii×p. grandidentata. Trees 2012, 26, 743-750.
[7] Nakaba,S.; Arakawa,I.; Morimoto, H.; et al. Agatharesinol biosynthesis-related changes of ray parenchyma in sapwood sticks of Cryptomeria japonica during cell death. Planta 2016, 243, 1225-1236.
[8] Obara, K.; Kuriyama, H.; Fukuda, H. Direct evidence of active and rapid nuclear degradation triggered by vacuole rupture during programmed cell death in zinnia1. Plant Physiology 2001,125, 615-626.
3、other comments is marked in the attached file.
Re:We are very grateful to your all comments on the manuscript. After discussion by all our authors, we accept all your suggestions and made careful modifications to original manuscript (in the attached file).

Reviewer 2 Report
This manuscript describes development and evaluation of methodology to measure specific flavenoids and terpenoids in wood. The reasons, that such methods are necessary to study heartwood formation, and the methodology and its evaluation, are described clearly and with mostly enough detail (see some minor comments).
A statement about reproducibility of spectra from replicate extractions of different wood samples from each region should be included somewhere obvious. Information on variability and reproducibility of spectra would be helpful.
Minor comments:
Line 123: State the source of standards trans-nerolidol, naringenin, isoliquiritigenin, dalbergenin and spinononetin
Line 145 was not were
Line 162 Define benzene alcohol solution (volume ratio 2:1). Which alcohol?
Line 164 How was the material milled? Name the machine? Temperature?
Line 166 aqueous solution of dioxane was extracted three times. How exactly was this done? Volumes, what solvents?
The Results and Discussion section needs to split into more paragraphs to help readers.
The title of Figure 2 should name the compounds in the order they appear in the graphs, and the names on horizontal axes of the graphs should be the same as in the title. State which growth ring is oldest - 1 or 10 - in the graphs or the Figure legend. Could use these numbers in the Results & Discussion text when referring to regions of the wood. This is done in lines 296, 299 but would be helpful earlier.
In legend to Figures 3 and 4 state if the colours and numbers refer to the samples/growth rings. Again indicate which is inner and which outermost.
In legend to Figure 5 indicate which numbered ring is inner and which outermost.
Author Response
Thank you for your suggestions on the grammar, experimental process and image quality of the manuscript. We think your comments are pertinent and valuable, which is very helpful to improve the quality of our manuscripts. Therefore, we accept all your suggestions after dissusion.
This manuscript describes development and evaluation of methodology to measure specific flavonoids and terpenoids in wood. The reasons, that such methods are necessary to study heartwood formation, and the methodology and its evaluation, are described clearly and with mostly enough detail (see some minor comments).
A statement about reproducibility of spectra from replicate extractions of different wood samples from each region should be included somewhere obvious. Information on variability and reproducibility of spectra would be helpful.
Re: As the information provided by UV and IR spectra is the structural change or relative content change of components, it is different from the absolute quantitative method of GC/MS and LC/MS (methodology needs to be established). At the same time, in order to highlight the purpose of exploring the radial changes of lignin structure and content in this section, we have not described much about this method. However, as you suggested, it is important to add the necessary repetitive expressions. We have added statement about the repeatability of the spectral results in manuscript in P5 line 190, P5 line 197, P8 line 304 and P8 line 327.
Minor comments:
Line 123: State the source of standards trans-nerolidol, naringenin, isoliquiritigenin, dalbergenin and spinononetin
Re: Thank you! We have added key information in P3, line 26.
Correct: Appropriate amounts of trans-nerolidol, naringenin, isoliquiritigenin, dalbergenin, and spinononetin (Shanghai Yuanye Biotechnology Co., Ltd, Shanghai, China) were dissolved in methanol, and the volume was fixed at 10 mL. The purities of all standards were above 98%.
Line 145 was not were
Re: We've corrected grammar error.
Correct: The trans-nerolidol was analyzed by Agilent 5975C GC-MS (Agilent Technologies Inc., USA) equipped with HP-5 column (30m×0.25 mm,0.25μm film thickness).
Line 162 Define benzene alcohol solution (volume ratio 2:1). Which alcohol?
Re: Thank you for your careful checks again!
Correct: First, dried wood flour (4.0000 g) was accurately weighed and extracted with benzene/ethanol mixed solution (volume ratio 2:1) in a Soxhlet extractor until the extract solution was colorless.
Line 164 How was the material milled? Name the machine? Temperature?
Re: We have added the informaton about mechine and temperature.
Correct: Thereafter, the solution was dried at low temperature (50℃) and subsequently milled by planetary ball mill (Hunan Fukasi Experimental Instrument Co., Ltd, China) for 4 h.
Line 166 aqueous solution of dioxane was extracted three times. How exactly was this done? Volumes, what solvents?
Re: Thank you for your question. It is that we are not clear about it, and we have adopted a more convenient expression for readers to understand.
Correct: Next, 40 ml of dioxane aqueous solution (water: dioxane = 4:96) was added. After magnetic stirring in the dark for 24 h, the supernatant was collected through centrifugation. Repeat the above operation three times. After filtration, the extracts were combined.
The Results and Discussion section needs to split into more paragraphs to help readers.
Re: We have made some adjustment.
The title of Figure 2 should name the compounds in the order they appear in the graphs, and the names on horizontal axes of the graphs should be the same as in the title. State which growth ring is oldest - 1 or 10 - in the graphs or the Figure legend. Could use these numbers in the Results & Discussion text when referring to regions of the wood. This is done in lines 296, 299 but would be helpful earlier.
In legend to Figures 3 and 4 state if the colours and numbers refer to the samples/growth rings. Again indicate which is inner and which outermost.
In legend to Figure 5 indicate which numbered ring is inner and which outermost.
Re: Thank you for this your careful checks! Your meaningful suggestions will helpful to improve our manuscript. In addition, we have added the numbers of regions in manuscript earlier (line 248)
(1) We have adjusted the order of the compounds in Fig.2.
Correct: Figure 2 Changes in the radial distribution of naringenin, isoliquiritigenin, dalbergenin, formononetin and trans-nerolidol. The horizontal axis shows the change of the growth ring from the pith (the first growth ring) to the outermost sapwood (the tenth growth ring). heartwood (HW; 1, 2, 3), transition zone (TZ; 4X, 4B), and sapwood (SW; 5, 6, 7, 8, 9, 10).
(2) Correct:
Figure 3. UV-Visible spectra lignin samples from different growth ring. The growth ring from the pith (the first growth ring) to the outermost sapwood (the tenth growth ring) were distinguished by different colors. Heartwood (HW; 1, 2, 3), transition zone (TZ; 4X, 4B), and sapwood (SW; 5, 6, 7, 8, 9, 10).
Figure 4. IR spectra of lignin samples from different growth ring. The growth ring from the pith (the first growth ring) to the outermost sapwood (the tenth growth ring) were distinguished by different colors. Heartwood (HW; 1, 2, 3), transition zone (TZ; 4X, 4B), and sapwood (SW; 5, 6, 7, 8, 9, 10).
(3) Correct:
Figure 5. Radial variation of mineral elements (Ca, Mg, Fe, Sr, K, Zn,Cu, Na) along the growth ring. Mineral elements are marked in the upper right corner. The horizontal axis shows the change of the growth ring from the pith (the first growth ring) to the outermost sapwood (the tenth growth ring). heartwood (HW; 1, 2, 3), transition zone (TZ; 4X, 4B), and sapwood (SW; 5, 6, 7, 8, 9, 10).
Reviewer 3 Report
please see the file atteached

Author Response
Thank you for your comments. We think your suggestions are pertinent and helpful to the improvement of manuscript quality. After discussion, we accept all your suggestions (detailled reply in the attached file).
